# Development of a Novel Interactive Multimedia E-Learning Model to Enhance Clinical Competency Training and Quality of Care among Medical Students

**DOI:** 10.3390/healthcare8040500

**Published:** 2020-11-20

**Authors:** Yu-Ting Hsiao, Hsuan-Yin Liu, Chih-Cheng Hsiao

**Affiliations:** 1Department of Ophthalmology, Kaohsiung Chang Gung Memorial Hospital, Kaohsiung 83301, Taiwan; yuting1008@cgmh.org.tw; 2Department of Medical Education, Kaohsiung Chang Gung Memorial Hospital, Kaohsiung 83301, Taiwan; liu.hsuanyin@gmail.com; 3Division of Hematology/Oncology, Department of Pediatrics, Kaohsiung Chang Gung Memorial Hospital and Chang Gung University College of Medicine, Kaohsiung 83301, Taiwan; 4Chang Gung Memorial Hospital Research Centre for Medical Education, Taoyuan 333, Taiwan

**Keywords:** eBooks, medical education, clinical competencies, interactive, e-learning

## Abstract

Clinical competencies consisting of skills, knowledge, and communication techniques should be acquired by all medical graduates to optimize healthcare quality. However, transitioning from observation to hands-on learning in clinical competencies poses a challenge to medical students. The aim of this study is to evaluate the impact of a novel interactive multimedia eBook curriculum in clinical competency training. Ninety-six medical students were recruited. Students in the control group (*n* = 46) were taught clinical competencies via conventional teaching, while students in the experimental group (*n* = 50) were taught with conventional teaching plus interactive multimedia eBooks. The outcomes of clinical competencies were evaluated using Objective Structured Clinical Examination (OSCE) scores, and feedback on their interactive eBook experiences was obtained. In the experimental group, the average National OSCE scores were not only higher than the control group (214.8 vs. 206.5, *p* < 0.001), but also showed a quicker improvement when comparing between three consecutive mock OSCEs (*p* < 0.001). In response to open-ended questions, participants emphasized the importance of eBooks in improving their abilities and self-confidence when dealing with ‘difficult’ patients. Implementing interactive multimedia eBooks could prompt a more rapid improvement in clinical skill performance to provide safer healthcare, indicating the potential of our innovative module in enhancing clinical competencies.

## 1. Introduction

The two essential bonds in health care and medical education—doctor–patient and teacher–learner relationships—have been at the core of medicine for decades. As patients rightfully trust the systems to adequately train doctors that are caring for them, the focus on establishing patient safety in medical education can provide confirmation of that trust, and therefore is of interest among medical educators. Furthermore, e-learning has become increasingly utilized in medical education globally, and educators must embrace the need for strategic improvement if knowledge for healthcare safety is to find its way to their students [1]. As many rapid advances in online medical educational systems have been made, there are several e-learning resources to provide different strategies for medical students to enhance their knowledge and performance [2,3,4]. Therefore, the question of its effectiveness and its integration into the medical education curriculum is a matter of importance [5,6,7].

Emphasis on patient safety and high-value care necessitates the acquisition of clinical skills, knowledge and attitudes as competencies in medical practice. During the final years of study, medical students are required to do rotations at hospitals and clinics, and play an active role in patient care. The concept of competencies is important as it implies a developmental transition from a medical student to, ultimately, a skilled and expert practitioner. To evaluate the complex notion of clinical competence, the Objective Structured Clinical Examination (OSCE) has developed into a mainstream assessment method in the education and licensing of physicians. OSCE testing has quickly become the standard process of certification, and is a part of the Medical Licensing Examination for medical graduates in various regions, including the U.S., Canada, the United Kingdom, and Taiwan. In 2011, the Taiwan Medical College Accreditation Council established 81 clinical competencies that should be acquired upon graduating. The clinical competencies included fields in physical examination skills, image interpretations, laboratory diagnosis techniques, procedure skills, treatment techniques and other activities [8,9]. Although the learning objectives are clearly defined, there is still a lack of well-organized curriculums to teach the clinical competencies, which in turn could result in inadequate clinical skills and poor doctor–patient relationships. A study has reported that 55.9% of students were confident that they had acquired the clinical skills required to become a resident, and 70.7% were satisfied with the quality of their medical education [10]. As such, in order to acquire sufficient knowledge and clinical skills, medical students have to spend more time self-studying clinical skills. Medical schools may also hold small sets of OSCE testing to help students in clinical competency training and to also pass the required assessment. However, the labor- and resource-intensive nature of OSCE testing makes it difficult for most medical schools to consistently invest this amount of time and money [11]. For example, Quebec Medical College reported spending USD 1080 for each student in a comprehensive OSCE [12]. Therefore, not only is it important to integrate clinical competencies into a new curriculum to help students, but also to provide students with suitable self-studying learning resources to optimize healthcare quality and patient safety.

Thanks to the advances in network technology, medical students are able to enhance their knowledge on patient care by e-learning nowadays. Ekenze et al. reported that most medical students are familiar with Internet tools and use them for learning, and they believe that the tools may be useful in integrating e-learning modality in the traditional mode of surgical education [13]. In fact, administrators and learners both find that e-learning enhances the teaching and learning experience [2]. E-learning is more efficient in allowing learners to gain knowledge and skills faster than traditional instructor-led methods [2,14]. In turn, this efficiency may be converted into improved motivation for learning. Evidence further suggests that e-learners have demonstrated increased retention rates and better utilization of content, resulting in better learning outcomes [14]. Furthermore, e-learners can select from a large variety of multimedia designs, including interactivity, feedback, and practice exercises to accommodate their different learning styles. This advantage in e-learning improves content accessibility, provides a personalized learning experience, further ensuring students are equipped with the knowledge, skills and attitudes necessary to function safely [15,16].

Despite the many benefits of e-learning, one problem is often faced: e-learning generally provides passive ways of learning, which lacks interactive instructional strategies and one-on-one guidance [3,17]. It is acknowledged that interactive learning offers a stronger learning reinforcement and helps to maintain the learner’s interest [2]. For medical students to develop clinical competencies, they should learn by relating acquired knowledge to clinical experiences, and further applying these skills in daily practice [2,18]. Therefore, classes in empathy, medical knowledge, patient care, clinical skills, and communication are integrated into the curriculum in medical schools, and the use of simulated models or standardized patients places theory in a clinical scenario [19]. However, at present, there are no practical interactive, self-studying tools for strengthening clinical competencies.

In this study, an eBook editing software (SimMAGIC eBook, Hamastar technology company, Taiwan) with simple and highly customizable interface was employed to create 81 interactive multimedia eBooks for training and learning clinical competencies. This eBook software has a wide array of editing functions and enables editors to produce a simulative eBook by integrating various forms of multimedia into each eBook. Students are required to take simulated tests by interacting with the content in eBooks, and initial feedback could be given immediately to enhance the learning experience. A learning-management system was also designed to track the students’ outcome assessments [20]. The aim of this study is to investigate the effect of interactive eBooks on the enhancement of clinical competencies, and their impact on the students’ OSCE performance. The second objective of this study is to provide a further understanding of how the interactive eBooks are perceived by medical students in practical healthcare settings.

## 2. Materials and Methods

### 2.1. Study Design and Participants

The participants enrolled in this study were medical students who were in their final year of medical school and doing clinical rotations in Kaohsiung Chang Gung Memorial Hospital, Taiwan. The control group consisted of 46 students who were taught clinical skills via conventional teaching, which consisted of teaching through direct patient care by a group of clinical teachers and senior physicians. The experimental group consisted of 50 students, which were taught by conventional teaching plus interactive multimedia eBooks. The learning outcomes of clinical competencies were evaluated using mock OSCEs and OSCE scores in the Taiwan’s National Medical Licensing Examination (National OSCE) to measure the clinical skill performance. Three mock OSCEs were held by Kaohsiung Chang Gung Memorial Hospital in August, November, and February. In the end, their final performance of clinical competencies was evaluated by the National OSCE. For an in-depth understanding of student experience with the eBooks, students in the experimental group were asked to perform a pre- and post-test included in each eBook session and anonymously answer a Likert-scale questionnaire. The experimental timeline is illustrated in Figure 1.

In this study, we used SimMAGIC eBook software to create 81 interactive multimedia eBooks for learning clinical skills. SimMAGIC eBook editing software was introduced in a previous study [20], allowing editors to use various interactive functions such as drag and drop, filling in the blanks, matching and sorting exercises, pop-ups, animations, and clips of audio and videos to create eBooks (Figure 2). The saved eBook can be edited or updated anytime, and anyone can use the eBook software. In this program, at first, we collected the PowerPoint slides of clinical competencies from clinical teachers. The contents were classified into different fields including physical examination (31 topics), visual image interpretation (6 topics), laboratory examination (8 topics), procedure skills (16 topics), therapeutic skills (14 topics) and others (6 topics). The contents were reviewed by other clinical teachers, who made sure the contents followed and covered the learning objectives.

Next, we integrated multimedia resources such as PowerPoint text, images, video, and audio into each interactive multimedia eBook, and emphasis was given to transitioning passive and instructor-led lectures into simulative and interactive educational materials (Figure 3). Afterwards, we uploaded the newly created eBooks to a Bookshelf Management Cloud Platform, which was compatible with eBook file formats. The eBooks can be downloaded to PC, tablets and mobile phones anytime and anywhere, permitting increased accessibility. A learning-management system is available on the platform, allowing medical educators to track the learners’ accession of each module and their achievement of competencies. The Bookshelf Management Cloud Platform enables offline reading, and the offline user history is recorded and uploaded to the platform upon connection to the Internet.

In both groups, learning outcomes of clinical competencies were accessed by holding 3 mock OSCEs to measure the learners’ clinical skill performance; the first mock OSCE was conducted on participants prior to commencement of the study, the second was arranged during the study period, and the third mock OSCE was held before the approaching National OSCE. National OSCE scores were also taken into account in both the control and experimental groups. In the experimental group, besides participants being asked to complete a pre-test and a post-test which were included in each eBook learning session, the students also took a feedback questionnaire on eBook satisfaction and its impact in healthcare practice near the end of the study period. Each student was given a different username and password for the eBook module in order to track their learning process individually.

### 2.2. Data Collection

We modified the Kirkpatrick model to evaluate the outcome of our interactive multimedia eBooks using two major approaches: learning outcomes and self-reported measures in confidence and dealing with problems of patients [21]. Learning outcomes were assessed by average scores of pre-test and post-test and cognitive learning gain. The short tests consisted of 5 multiple-choice questions prepared by the educator, and they were given to the students before and after participating in our eBook module. We used the pre/post-test to determine cognitive gain during eBook learning. According to Hake’s criteria for effectiveness of educational intervention, absolute learning gain (%post-test score − %pre-test score) and class-average normalized gain (%post-test score − %pre-test score) / (100 − %pre-test score) were also calculated. A class-average normalized gain of 30 % was considered significant [22].

A 19-item questionnaire was designed to investigate the students’ reactions to eBooks. The first part of the questionnaire consisted of 8 items regarding the quality of teaching material and their attitudes towards how helpful the eBooks were when they were performing clinical competencies in practical settings. The second part involved 5 items, which assessed the participants’ perception in raising confidence during clinical practice and in developing good doctor–patient relationships. The last part of the questionnaire contained 6 items regarding the eBook platform user interface and the manner of eBook interactive content delivery. Participants rated their responses on a five-point Likert scale, with 5 being strongly agree and 1 being strongly disagree. Additional qualitative data were collected from written responses from open-ended questions on the impact of eBooks on their practice at the end of the self-assessment questionnaire.

### 2.3. Data Analysis

We compared the pre/post-test data and National OSCE scores using Student t-test. The ANOVA test was used to determine significant differences in mock OSCE scores, and Fisher’s least significant difference (LSD) was used for post hoc comparisons. A *p* value of < 0.05 was considered statistically significant. All statistical analyses were conducted using SPSS version 20 (SPSS Inc., Chicago, IL, USA).

## 3. Results

Ninety-six medical students performing clinical rotations took part in this study. Twenty-three percent of the participants were female and 77% were male. Through the three mock OSCEs held in the study period, both groups showed significant improvement (control group, *p* = 0.006; experimental group, *p* < 0.001) (Table 1). The average scores of the 46 participants in the control group showed a significant difference when comparing the third mock OSCE to the first one (*p* < 0.001) (Figure 4). In the experimental group, a significant improvement was found between the second and first mock OSCE (*p* < 0.001), and the third and the first mock OSCE scores (*p* < 0.001) (Figure 4). Last but not least, all of the participants in our study passed the National OSCE; the average OSCE scores in the experimental group were higher than the scores in the control group (214.8 in the experimental group vs. 206.5 in the control group, *p* < 0.001) (Figure 5). In addition, in the experimental group, there was a significant difference in improvement when comparing between the pre-test and post-test scores (74.8 ± 9.87 for pre-test vs. 86.6 ± 11.49 for post-test, *p* < 0.001) (Figure 6). The absolute learning gain was 11.8, whereas the class-average normalized gain was 46.8%.

In the experimental group, all of the 50 participants returned the feedback questionnaire (response rate 100%). The average rating for the majority of the questions ranged between 4.0 and 4.5 (Table 2, Table 3 and Table 4), suggesting that a majority of participants perceived interactive multimedia eBooks as being useful in enhancing clinical competencies. Table 2 shows the students’ response regarding the quality of teaching material and their attitudes towards how clinically relevant the eBooks were. Table 3 presents participants’ responses in confidence boosting during clinical practice and in helping to develop good doctor–patient relationships after using eBooks. Table 4 shows the students’ responses regarding the eBook platform user interface.

In response to the open-ended questions, participants pointed out the benefits of eBook learning in improving their confidence in practicing clinical competencies, especially in ‘difficult’ patients, having more empathy by performing clinical competency skills with ease, and allowing them to be able to read subtle cues from patients, and have different solutions to different scenarios to build a stronger doctor–patient relationship (Table 5). Most of the negative aspects of the students’ responses were related to the functional design of the platform. Several participants also commented on the limitation of downloading only one eBook at a time, the lack of systematic organization with which eBooks are arranged and ordered, and the need for customized labels for marking eBooks as read or unread.

The eBooks used in our study were accessible on mobile devices. In terms of how the participants accessed the eBooks, 46% of the participants reported that they used mobile devices as a primary device for accessing the eBooks, while 54% used a desktop computer or a laptop. Of the students who obtained eBooks from mobile devices, about 61% used tablets, 26% used smartphones with an Android system, and 13% with an iPhone.

## 4. Discussion

Efforts to discern patient safety and quality of care constantly push medical education into new territory [1]. As new e-learning environments are introduced into the field of medical education, there is an increasing need for establishing evaluations for these innovative e-learning systems. In concordance with other previous studies [3,23,24,25], the present study shows that students have positive perceptions upon integrating a new e-learning module to supplement traditional training materials. Our findings indicate that interactive multimedia eBook training offers a promising approach to enhance students’ knowledge and skills on improving clinical competencies to promote safety and reliability in healthcare.

Strategic improvement in health and healthcare starts with focusing on improving medical education. In the continuum of medical education, the beginner learns by applying a defined set of rules to new unforeseen situations, and as learners acquire experience, they become more intuitive as rule-bound behaviors begin to emerge [1]. During this process of training future licensed physicians, effective and safe clinical systems are required [26]. We believe that interactive multimedia eBooks represent a novel and unique contribution in the field of e-learning in training clinical competencies. In the past, clinical competencies were taught by observations and through traditional lectures in clinical clerkship [9]. Upon clinical rotations, transitioning from clerkship observation to hands-on learning through patient care entailed challenges for medical students nearing entry to practice [10,27]. The participants in the experimental group acknowledged that interactive multimedia eBooks allowed them to be more confident and efficient in performing clinical competencies, and helped them to establish a good doctor–patient relationship during clinical practice.

The present study indicates a successful implementation of using e-learning to support clinical competency training in medical students. To our knowledge, this is the first eBook learning module in which students can acquire clinical competency skills. Our data show that integrating eBooks into clinical training could result in a more rapid significant increase in clinical skill performance upon comparing the results between the three mock OSCEs within both groups. Furthermore, our study suggests that there is a significant difference in the learning outcomes of the pre-test and post-test in the experimental group. Most training resources supporting clinical education consist of video demonstrations of clinical skills and/or mock OSCE exams [7,28]. Although the mock OSCEs show positive effects, the cost of holding formative mock OSCEs on a regular basis is high as practical considerations include training of examinees and standardized patients, test development costs, and the maintenance of facilities to administer the test [9,29].

Evidence has shown that when compared with traditional instructor-led teaching methods, e-learning can result in significant reduced costs, sometimes as much as 50% [29]. The current medical education curriculum and tuition includes clinical rotations and face-to-face teaching, especially in the final years of medical school. Defining the cost of face-to-face teaching is complex [8,9]. Our eBook module is a technology to deliver another training technique, making assessing clinical experiences simpler and more efficient, rather than replacing clinical training. Furthermore, medical educators can update the latest knowledge and skills, and add new topics any time after the original module is established. The costs of eBooks may be difficult to estimate as it may benefit several years of medical students and in different institutions. We believe our eBook module offers benefits for learners and educators, delivering cost-effective, repeatable, and standardized clinical training. Future research is recommended to inform whether eBooks can serve as an adequate supplement for other OSCE training modules on the basis of cost-effectiveness.

Our findings indicate that the mobile learning environment plays a crucial role by offering flexibility. As medical students spend a significant amount of time outside clinical training programs or classrooms, providing increased accessibility anytime and anywhere can allow them to benefit from a seamless learning environment [7,30]. Furthermore, educators can update the eBook in a timely manner to ensure delivery of the latest evidence-based medical knowledge [2]. Our eBook module can allow the creation of certain simulation curricula to meet specific needs, as teaching with real patients are limited to the disease they present with [31]. To further enhance the students’ learning experience, blended learning program developments are needed on how to integrate interactive multimedia eBook resources into their teaching.

Previously, despite the many benefits of e-learning in medical education, the major barrier is a lack of interaction [7]. The keys to overcoming this challenge in our eBook module were identifying what sort of problems the students would face beforehand, providing timely and specific feedback, and setting up an online communication channel for students and educators. Besides integrating interactive multimedia designs in our eBooks, we also recruited experienced clinical educators to point out the problems that medical students often encounter or should improve on when practicing the clinical competencies in clinical scenarios. Emphasis was then given on mapping out the suitable interactive multimedia design and the responses of the automated feedback function of that particular topic. Students can use the online communication tool in the eBook platform to post questions that were still unanswered after completing the eBook to educators, enabling students to receive personalized feedback for self-improvement. After answering the students’ questions, educators can not only have a better grasp of the students’ training process, but also make improvements on future versions of the eBooks. Simulation with interactive virtual reality (VR) has also emerged as a new method in which we can deliver medical education. However, there are difficulties with introducing any new technology, as it requires faculty space and support [31].

An additional strength of our Bookshelf platform is that it has automated tracking functions. The tracking function can record the learners’ activities and assessment outcomes during both offline and online conditions, and report them back to educators. Although several administrative functions have been facilitated, the learners have suggested some modifications in the user interface. Some participants commented on the limitation of downloading only one eBook at a time, the lack of systematic organization with which eBooks are arranged and ordered, and the need for labels marking eBooks as read or unread. These findings indicate that the user interface should be more user-friendly and updated constantly to benefit the students’ learning experience.

We acknowledge some limitations in our study. First, we only conducted and evaluated the use of interactive multimedia eBooks at a single institution. Thus, our sample size was relatively small. Second, the effectiveness of interactive multimedia eBooks on OSCE scores remains modest. Although our data showed significant improvement after eBook intervention, other medical learning resources may also be involved in OSCE scores. Therefore, in future studies, the eBook learning setting could be extended to several years of medical students or to other medical curriculums or institutions to obtain generalizable results.

## 5. Conclusions

Application of our interactive multimedia eBook was associated with a more rapid improvement in clinical skill performance and enhanced the students’ abilities and self-confidence when dealing with difficult patient encounter settings. To our knowledge, this is the first eBook in the form of interactive and multimedia design that has been established for clinical competencies. Further learning program development should be established to effectively integrate the eBooks into other areas in the medical curriculum to provide safer and better healthcare.

## Figures and Tables

**Figure 1 healthcare-08-00500-f001:**
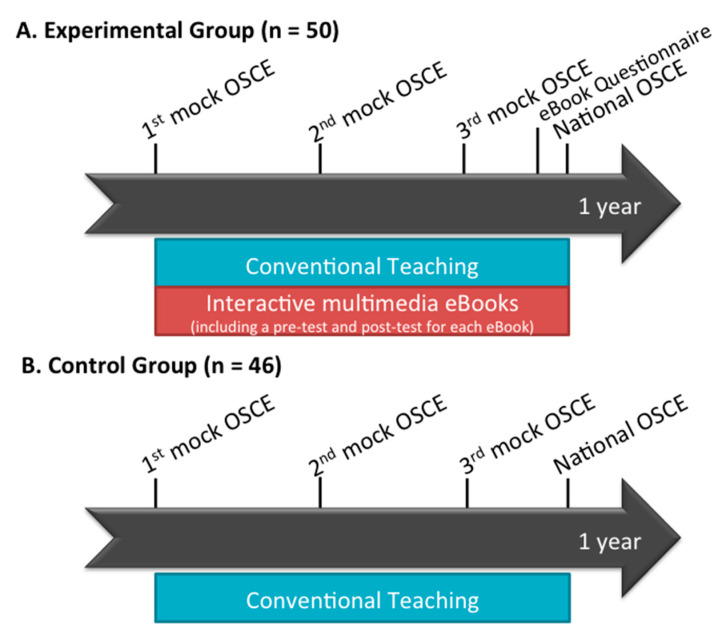
Schematic illustration of the experimental timeline.

**Figure 2 healthcare-08-00500-f002:**
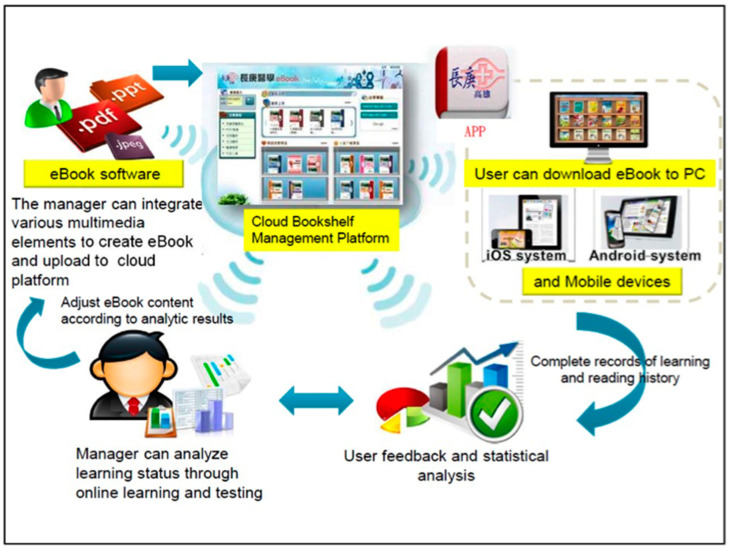
Establishment of the Bookshelf Management Cloud Platform.

**Figure 3 healthcare-08-00500-f003:**
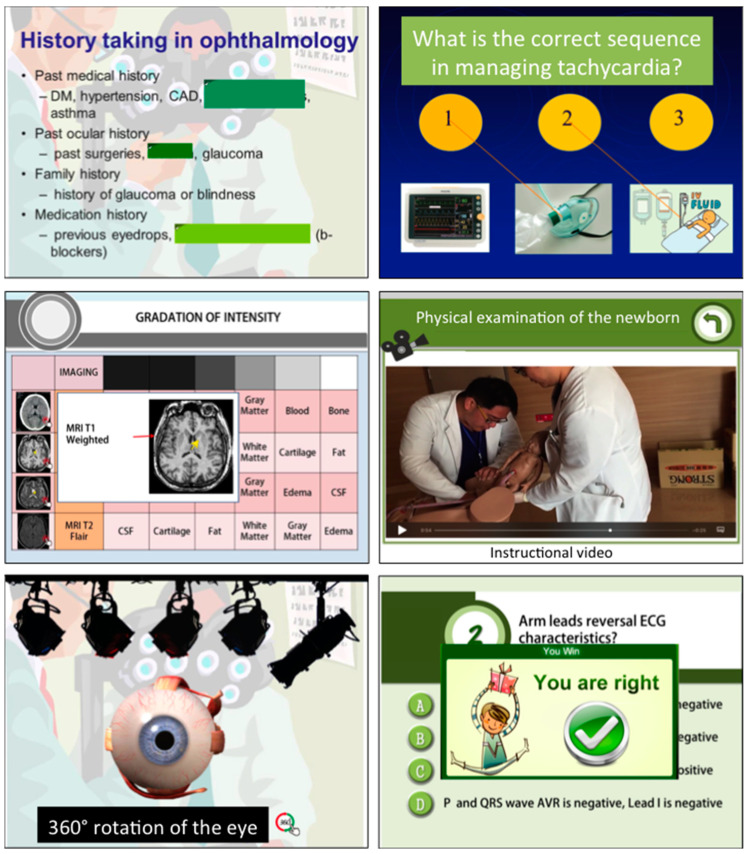
Interface of various interactive functions in our eBook module, including simulation models in anatomy, instructional videos, labeling exercises, comprehensive multi-page assessments, and interactive scenarios with instructive feedback.

**Figure 4 healthcare-08-00500-f004:**
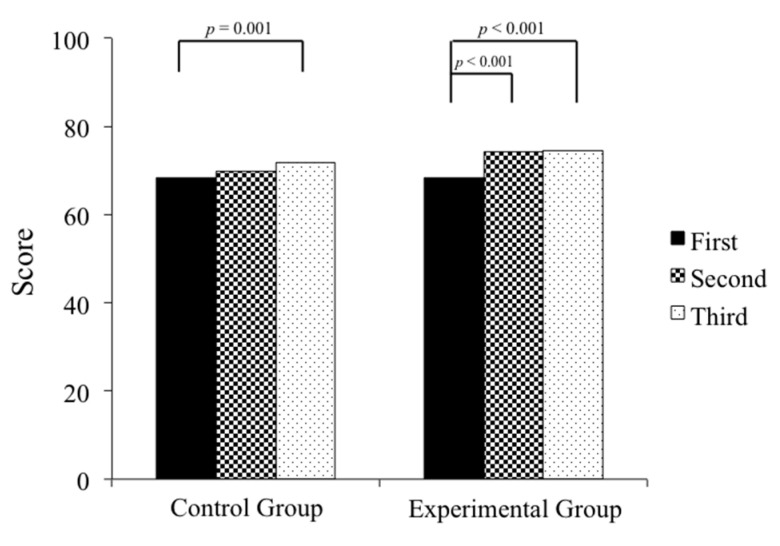
The average mock OSCE scores of the control group showed a significant difference when comparing the third exam to the first one (*p* = 0.001), whereas in the experimental group, there was already a significant improvement found between the second and first, and the third and the first mock OSCE scores (*p* < 0.001).

**Figure 5 healthcare-08-00500-f005:**
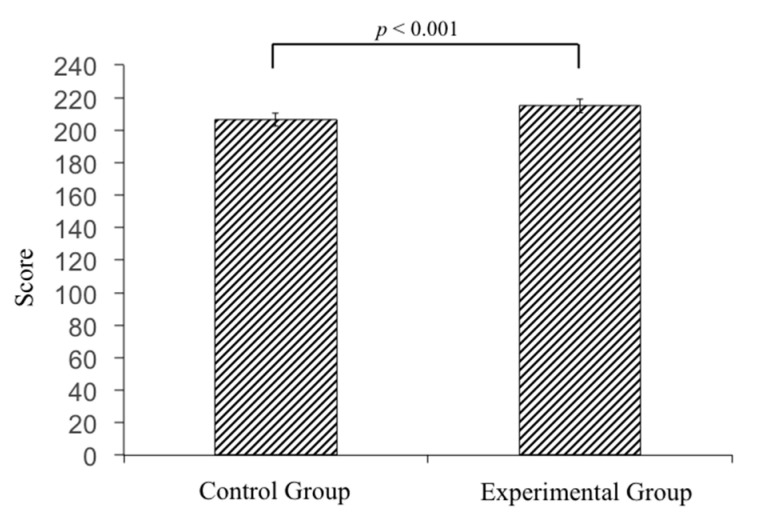
The average scores of National OSCE were higher in the experimental group than the scores in the control group (214.8 in the experimental group vs. 206.5 in the control group, *p* < 0.001).

**Figure 6 healthcare-08-00500-f006:**
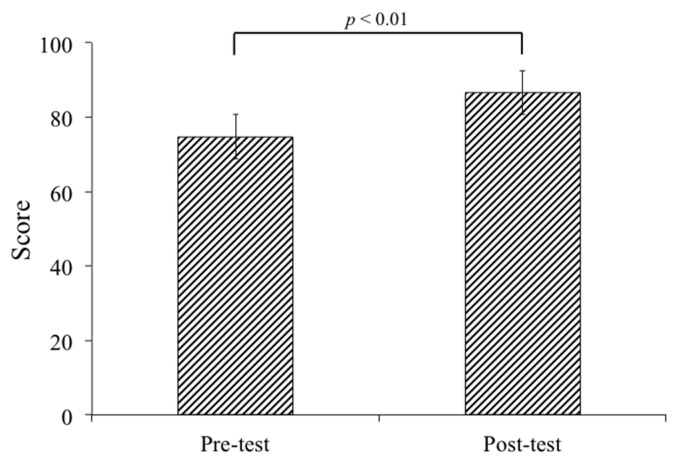
The students’ performance score on the post-test was significantly higher than the pretest (74.8 ± 9.87 for pretest vs. 86.6 ± 11.49 for posttest, *p* < 0.001) in the experimental group. Student’s t-test was used in comparison between pre-test and post-test. A *p*-value < 0.05 is considered statistically significant.

**Table 1 healthcare-08-00500-t001:** Mock Objective Structured Clinical Examination (OSCE) mean scores of the control group and the experimental group.

	Mean Scores	*p* Value
Control group		0.006
First	68.26 ± 6.06	
Second	69.74 ± 4.57	
Third	71.78 ± 4.79	
Experimental group		<0.001
First	68.30 ± 6.40	
Second	74.14 ± 5.47	
Third	74.60 ± 6.99	

**Table 2 healthcare-08-00500-t002:** Students’ responses on the quality of teaching materials and how helpful the eBooks were when they were performing clinical competencies in practical settings. (*n* = 50).

No.	Questionnaire	Average Rating
1	Interactive materials improve my learning efficiency	4.22/agree
2	Interactive materials can improve the retention of knowledge	4.23/agree
3	Interactive materials increase my focus in learning	4.03/agree
4	Interactive materials enhance learning motivations	4.12/agree
5	Teaching materials improve my ability in solving problems	4.09/agree
6	The teaching material helps to improve my clinical skills	4.22/agree
7	The teaching material improves my ability in making appropriate treatment decisions based upon patients’ needs	4.22/agree
8	eBooks increase my interest in dealing with clinical competency aspects of patient care	4.20/agree
Overall average		4.17/agree

A five-point Likert scale was used, where rating 1: Strongly Disagree, 2: Disagree, 3: Neutral, 4: Agree, 5: Strongly Agree.

**Table 3 healthcare-08-00500-t003:** Students’ responses in raising confidence during clinical practice and in developing good doctor-patient relationships. (*n* = 50).

No.	Questionnaire	Average Rating
1	The teaching materials raise my confidence in handling difficult patient encounters	4.32/agree
2	Self-studying eBooks could enhance my skills to promote better doctor-patient relationships	4.32/agree
3	The teaching contents make it easier to understand and interpret information patients are giving me	4.31/agree
4	eBooks can allow me to obtain information from patients in a systematic way	4.32/agree
5	eBooks allow me to perform my clinical competency skills with ease, making me more aware of how patients react to me	4.29/agree
Overall average		4.31/agree

A five-point Likert scale was used, where 1: Strongly Disagree, 2: Disagree, 3: Neutral, 4: Agree, 5: Strongly Agree.

**Table 4 healthcare-08-00500-t004:** Students’ responses regarding the eBook platform user interface. (*n* = 50).

No.	Questionnaire	Average Rating
1	Easy-to-use eBook platform user interface	4.11/agree
2	eBook platform features a variety of useful functions	4.18/agree
3	The platform enhances my learning experience	4.03/agree
4	The eBook reading interface is easy to understand	4.08/agree
5	The functions in reading interface facilitate learning	3.95/neutral
6	Using eBook platform can increase my interest in learning	4.15/agree
Overall average		4.08/agree

A five-point Likert scale was used, where rating 1: Strongly Disagree, 2: Disagree, 3: Neutral, 4: Agree, 5: Strongly Agree.

**Table 5 healthcare-08-00500-t005:** Students’ open-ended responses from eBook learning.

Category	Responses
Confidence in handling difficult clinical settings	■ Helped me deal with difficult patients (e.g., difficult nasogastric tube insertion procedures) ■ Have a better understanding of what to expect of difficult patients ■ In a difficult patient setting, I could easily remember the alternative techniques and tips that were taught adequately in eBooks
More empathy towards the patient	■ To be able to perform clinical competency skills with more ease allows me to pick up on subtle cues from patients and recognize patients in distress ■ It helped me put myself in the patients’ shoes and become more considerate. ■ Be more empathic and see them from the patient’s point of view ■ Understand the patient’s concerns
Improvement in the doctor-patient relationship	■ Have different solutions when dealing with different difficulties, which is a foundation for a stronger patient-doctor relationship ■ Helped me control my emotions and uncertainty during practice due to sufficient training of competencies from eBooks

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
