# Peer review of "Development of a Novel Interactive Multimedia E-Learning Model to Enhance Clinical Competency Training and Quality of Care among Medical Students"

_healthcare, 2020, doi:10.3390/healthcare8040500_

Round 1

Reviewer 1 Report

An excellently written paper that addresses an important topic.  In the new world of COVID19, we will depend more and more on virtual learning of eTeaching.

The paper is easy to read and easily understandable.

A few minor details:  

  • Line 107, were not where.
  • Line 115,  the not their
  • Line 156-157: Not sure what the last sentence means.  Please expand the discussion.

There are similar aids to teaching, replacing live patients.  Two of these are simulations and the other is the use of virtual reality.  I suggest you might acknowledge that other methods exist.  Perhaps the combination of these methods would produce even better results.

I also suggest you add a discussion of costs.  What are your estimates of the cost of eBooks?  Your research was an add-on to teaching that all received.  Could eBooks be effective with reducing the more expensive face to face teaching?

One other point you might note.  Teaching with real patients are limited to the disease these patients present with.  Other diseases may not have an opportunity to be taught with real patients.

Over all, an excellent paper.

Reviewer 2 Report

This is an interesting paper giving details of a research project within medical education using a content management learning tool (eBook) alongside traditional methods of medical education as determined by the curriculum.  Such tools have been used for a long time in many educational settings, particularly in nursing so it is good to see that such use has been realised in medical education.  My concern rests with the two groups and the suggested findings.  The control group had the usual teaching experience and the experimental group appear to have received the usual teaching experience plus the eBook which even without the rigour of research would suggest that there should be a better response from the experimental group which indeed was the case.  As stated in the discussion (line 264) the use of such a platform contributes to the learning process but as the authors go on to state further research needs to be done in this area.
